# Geographic Range Overlap Rather than Phylogenetic Distance Explains Rabies Virus Transmission among Closely Related Bat Species

**DOI:** 10.3390/v14112399

**Published:** 2022-10-29

**Authors:** Maude Jacquot, Megan A. Wallace, Daniel G. Streicker, Roman Biek

**Affiliations:** 1School of Biodiversity, One Health and Veterinary Medicine, Boyd Orr Centre for Population and Ecosystem Health, College of Medical, Veterinary and Life Sciences, University of Glasgow, Glasgow G12 8QQ, UK; 2Centre for Virus Research, MRC-University of Glasgow, Glasgow G61 1QH, UK

**Keywords:** host shifts, cross-species transmission, rabies, genetic divergence, host-pathogen interaction, niche overlap, range overlap, *Myotis* bat, North America

## Abstract

The cross-species transmission (CST) of pathogens can have dramatic consequences, as highlighted by recent disease emergence events affecting human, animal and plant health. Understanding the ecological and evolutionary factors that increase the likelihood of disease agents infecting and establishing in a novel host is therefore an important research area. Previous work across different pathogens, including rabies virus (RABV), found that increased evolutionary distance between hosts reduces the frequency of cross-species transmission and of permanent host shifts. However, whether this effect of host relatedness still holds for transmission among recently diverged hosts is not well understood. We aimed to ask if high host relatedness can still increase the probability of a host shift between more recently diverged hosts, and the importance of this effect relative to ecological predictors. We first addressed this question by quantifying the CST frequency of RABV between North American bat species within the genus *Myotis*, using a multi-decade data set containing 128 nucleoprotein (N) RABV sequences from ten host species. We compared RABV CST frequency within *Myotis* to the rates of CST between nine genera of North American bat species. We then examined whether host relatedness or host range overlap better explains the frequency of CST seen between *Myotis* species. We found that at the within genus scale, host range overlap, rather than host relatedness best explains the frequency of CST events. Moreover, we found evidence of CST occurring among a higher proportion of species, and CST more frequently resulting in sustained transmission in the novel host in the *Myotis* dataset compared to the multi-genus dataset. Our results suggest that among recently diverged species, the ability to infect a novel host is no longer restricted by physiological barriers but instead is limited by physical contact. Our results improve predictions of where future CST events for RABV might occur and clarify the relationship between host divergence and pathogen emergence.

## 1. Introduction

The molecular biology of RNA viruses makes them prone to cross-species transmission (CST). These CST events result in either a dead-end infection with no onward transmission in the new host species (R_0_ = 0), a stuttering, unsuccessful chain of infections (0 < R_0_ < 1), or establishment in a novel host—a host shift (R_0_ > 1) [1,2]. RNA virus host shifts from animal reservoirs have spawned many emerging infectious diseases with large health and economic impacts, such as HIV/AIDs, H5N1 avian influenza and most recently, SARS-CoV-2 [3,4,5,6]. Between host species of different genera, evidence from both experimental and field data suggests that the probability of an RNA virus successfully establishing in a novel host increases with genetic relatedness between the donor and recipient host species [7,8].

However, what is unclear is whether this phylogenetic distance effect holds over different degrees of relatedness and whether it might become irrelevant at very fine taxonomic scales. There are reasons to doubt its relevance at both extremes of genetic distance. Among extremely divergent species all host shifts may become equally improbable, and conversely, among very closely related species, physiological differences among hosts may become relatively inconsequential. Thus, we do not yet know if high host relatedness still increases the probability of a host shift between more recently diverged hosts, and the importance of this effect relative to ecological predictors. This is important as, among mammals, the majority of CST is predicted to occur among closely related host species [9]. If reduced physiological barriers to sustained transmission mean that phylogenetic constraints are less relevant at this level, our ability to predict patterns of CST in nature at this scale might instead depend on alternative predictors.

Rabies virus (RABV) can infect a broad range of species, providing an opportunity to investigate what drives patterns of RNA virus host shifts in wild, multi-host communities [10]. RABV (genus: *Lyassavirus*, family: *Rhabdoviridae*) causes the zoonosis called rabies, a disease with one of the highest known fatality rates in humans [11]. The virus is capable of infecting all mammals, but paradoxically is maintained in distinct host species-associated transmission cycles, usually in carnivores or bats. RABV phylogenetic history is littered with evidence for CST [12], and its ability to shift hosts is demonstrated by the spread of the cosmopolitan dog rabies lineage into grey foxes, coyotes, and other wild mesocarnivores during European colonization of the Americas [13]. More recent CST events have created new reservoirs of sustained RABV transmission in species of mongoose and ferret-badgers [10]. This ability to establish in novel host species presents a significant danger to domestic animals and humans, and a challenge to RABV control [14].

The probability of RABV CST events initially occurring, as well as resulting in sustained host shifts, is influenced by both ecological and evolutionary factors [15,16] but disentangling their relative effects is challenging. Ecological factors’ major role in mediating RABV CST events was postulated after studies observed host shifts of RABV strains which were seemingly ‘pre-adapted’ to novel hosts, through standing genetic variation [12,17]. As a result, contact rates—within or between species—are essential to its onward transmission. Ecological factors, such as non-overlaps in host range, or differences in foraging or roosting niche, which reduce inter-species contact rates, could therefore be important in maintaining RABV variants in their host-specific cycles. However, for some host groups, the non-random clustering of RABV reservoirs on the phylogeny suggests that host-specific evolutionary factors also constrain RABV CST events [2,18]. The evolutionary history of RABV has also revealed a likely role of adaptive evolution in facilitating host shifts [12]. Indeed, it has been hypothesised that high genetic relatedness between closely related species facilitates CST due to the lower number of adaptive changes necessary for virus exchange [8].

Previous works have examined the importance of both ecology and evolution in determining the success of RABV host shifts across highly divergent bat species (nine genera and two of the four North American bat families) [8,19]. In these studies, higher host genetic relatedness was associated with a significantly higher frequency of both CST (external & internal nodes of the virus phylogeny) and host shifts (internal nodes only). Larger overlaps in species’ geographic range were also associated with significantly more frequent CST, though the strength of this effect was lower than that of genetic relatedness. In contrast, no association was detected between CST frequency and other ecological predictors in these data. However, it remains unknown whether host relatedness and range overlap, the primary determinants of CST frequency between highly divergent species, can still predict RABV CST between closely related species.

Here, our primary aim was to ask if high host relatedness can still increase the probability of a host shift between more recently diverged hosts, and the importance of this effect relative to ecological predictors. To do this, we estimate rates of RABV CST events over the virus’ phylogenetic history within a single bat genus (*Myotis)*, based on data from ten North American species collected over a 25-year period. *Myotis* (family: *Vespertilionidae*, subfamily: *Myotinae*) are small (most < 10 g), generally insectivorous bats with variable roosting habits and colony sizes, estimated to have diverged from other North American Vespertilionidae 20–25 MYA [20,21,22]. Rabies virus has emerged independently multiple times in the *Myotis* genus and a small proportion of variants form suspected multi-species transmission cycles out-with *Myotis* [8]. We firstly compare our estimates of CST frequency within *Myotis* to those for a previously published dataset containing nine bat genera [8], hypothesising that the lower average divergence among *Myotis* hosts presents fewer barriers to CST and host shifts in comparison. We then examine whether between-host genetic divergence, as opposed to range overlap or other ecological predictors, can best explain the rate of overall CST within *Myotis*, and events which led to successful host shifts. We predicted that compared to more divergent species, the *Myotis* RABV data would yield (*i*) higher CST rates overall; (*ii*) a higher proportion of CST events on internal branches, indicative of ongoing transmission in the new host, i.e., host shifts [19]; (*iii*) evidence for CST in a greater proportion of host species examined.

## 2. Material and Methods

### 2.1. Compilation of Myotis Associated RABV Sequences

We compiled a dataset of 128 RABV nucleoprotein (N) gene sequences (1350 bp) from *Myotis* bats collected across Canada and the USA. Most of these sequences (*n* = 112), were generated through a passive surveillance programme at the Canadian Food Inspection Agency’s Centre for Expertise for Rabies, between 1990 and 2015 [23]. RABV sequences used in the analysis were collected from individuals identified to *Myotis* host species through Cytochrome oxidase subunit 1 (CO1) barcoding. We only included RABV variants that fell within a monophyletic clade associated with the bat genus *Myotis* (MYCAN clade); strains normally associated with other bat genera were excluded from the analysis [23]. The collected sequences were supplemented by *Myotis* associated RABV sequences (*n* = 16) from the USA and Canada, available on Genbank as of May 2022.

The final dataset included RABV N gene sequences collected from ten *Myotis* host species: the South eastern bat *M. austroriparius* (*n* = 3), the California bat *M. californicus* (*n* = 24), the Eastern small footed bat *M. leibii* (*n* = 1), the Keens bat *M. keenii* (*n* = 8), the little brown bat *M. lucifugus* (*n* = 32), the Western long eared bat *M. evotis* (*n* = 28), the long-legged bat *M. volans* (*n* = 1), the Northern long-eared bat *M. septentrionalis* (*n* = 11), the western small footed bat *M. ciliolabrum* (*n* = 1) and the Yuma bat *M. yumanensis* (*n* = 19). The sequences used in this analysis can be found in Genbank under the accession numbers summarised in Appendix A.

### 2.2. CST Quantification within the Myotis Genus in Comparison to Bats of Multiple Genera

We modelled host species as a discrete trait for each RABV sequence over the genealogy by ancestral state inference using a discrete asymmetric phylogenetic diffusion model [24] in the ‘Bayesian Evolutionary Analysis Sampling Trees’ (BEAST) software v1.8.3 [25]. This approach estimates the probability of the internal nodes and branches being associated with a specific host, based on information about host states of the samples at the branch tips. A Bayesian stochastic search variable selection procedure [24] was employed to allow for CST between specific host pairs to be included or excluded from the model. Because CST events are likely to be rare, we quantified their frequency using a robust Markov jump (MJ) counting procedure that determines the posterior expectations of the number of CST along the branches of the tree [26]. For each host species pair, we divided the expected number of CST events (MJ count) by the sum of the tree branch lengths (in years) and by the combined number of sequences available for the two species composing the pair in order to obtain the mean number of CST events per year and per capita, hereafter referred to as the CST rate.

Analyses were performed under a general time reversible model of nucleotide substitution, with invariant sites, gamma distributed categories of rate variation and empirical base frequencies [27,28]. A lognormal relaxed molecular clock was used to calibrate the RABV phylogeny [29], along with a flexible Gaussian Markov Random Field (GMRF) coalescent model, the skyride [30]. A Markov chain Monte Carlo (MCMC) chain of 100 million steps was run and sub-sampled every 10,000 generations. The ‘Broad-platform Evolutionary Analysis General Likelihood Evaluator’ (BEAGLE) library was used to increase computational speed [25,31].

Whether a stationary distribution was reached, and sufficient mixing (i.e., effective sample size > 200) for all parameter estimates were checked in Tracer after removing the initial 10% of the samples as burn-in. Maximum clade credibility (MCC) trees were generated from tree output files from BEAST in TreeAnnotator v1.8.3 and annotated in the FigTree v1.4.3 graphical user interface (available at http://tree.bio.ed.ac.uk/software/figtree/, accessed on 27 October 2022).

BEAST log files were processed to obtain and visualise estimates of the number of inferred CST events between each host species pair over the complete history of the RABV phylogeny as well as the CST rate.

As a reference point for the amount of CST to expect between less closely related species, we also quantified the frequency of CST among North American bats of multiple genera by re-analysing the data from [8] using the same methodology as described above. This dataset consisted of 372 RABV N gene sequences from 17 bat species or species complexes. This includes species from the families *Vespertilionidae* and *Molossidae*, and the genera *Eptesicus* (*n* = 118), *Antrozous* (*n* = 3), *Lasiurus* (*n* = 121), *Lasionycteris* (*n* = 17), *Myotis* (*n* = 33), *Nycticeius* (*n* = 3), *Parastrellus* (*n* = 5), *Perimyotis* (*n* = 14), *and Tadarida* (*n* = 56) and 2 additional sequences not attributed to a host species. Among the *Myotis* genera within this data set, four discrete species states were defined: three of these represented single species (*M. californicus*, *n* = 10; *M. yumanensis*, *n* = 11; *M. austroriparius*, *n* = 2) whereas the fourth grouped sequences derived from *M. lucifugus* (*n* = 5), *M. evotis* (*n* = 4) and *M. thysanodes* (*n* = 1), which are thought to form a species complex. Here, two MCMC chains of 800 million steps were run and sub-sampled every 80,000 generations. Convergence to a stationary distribution was checked in Tracer after removing the initial 10% of the samples as burn-in.

### 2.3. Genetic and Ecological Host Predictors of CST within the Myotis Genus

For each *Myotis* host species pair, we considered four host variables, as well as sample size, as potential predictors of RABV CST events:

#### 2.3.1. Host Genetic Distance

We estimated the phylogenetic distance among host species (or host species clusters, see below) based on the average pairwise differences between mitochondrial CO1 sequences. In total, 171 CO1 sequences, representing all ten *Myotis* host species, were compiled from Genbank. Where possible, CO1 sequences were selected that matched the geographic origin of the RABV samples. Sequences selected were aligned using MUSCLE v3.8.31 [32,33]. A HKY + I + G model of nucleotide substitution was used to calculate the genetic divergence between CO1 sequences, selected using jModelTest v2.1.8 [28] that generated Akaike’s Information Criterion values corrected for small sample sizes, i.e., cAIC [27,28]. A distance matrix was calculated using the TREE PUZZLE analysis tool [34], and the R package ape [35].

#### 2.3.2. Range Overlap

Geographic range overlap was calculated as the size of overlapping area between the range of the each of the *Myotis* host species in the USA and Canada (Figure 1A). Range overlaps were calculated from NatureServe shape files (NatureServe 2016) using R packages maptools [36], spatstat [37], sp [38] and raster [39].

#### 2.3.3. Foraging Niche Difference

Three morphological measurements; wing aspect ratio (wing span^2^/wing area), wing loading capacity (body weight/wing and tail membrane area) and body length, were collected for each of the ten *Myotis* species from the literature [8,40,41] (Appendix A). Foraging niche difference was calculated as the mean Euclidean distance between these three morphological measurements, which can be used as a proxy for similarity in the foraging niche of bat species [8,19].

#### 2.3.4. Roosting Niche Overlap

Observational data on the summer roost types for the *Myotis* host species was gathered from the literature. For each species, we noted which of five types (tree hollows, man-made structures, cave, tree bark, or rock crevice) of summer roosting locations it was reported to use. The number of overlapping roost types (0–5) was used as a measure of similarity between the roosting niches of the species, and therefore their likelihood of co-roosting (Appendix A).

#### 2.3.5. Sample Size

Sample size, which can particularly impact the results for MJ, was included as an additional predictor. This predictor consisted of a symmetric matrix of the Manhattan distance between the number of RABV sequences representing each discrete species state [42].

Each predictor vector, originating from a converted matrix, was log-transformed and standardized using the R package gdata [43] to limit the impact of extreme values and to make variables as comparable as possible.

### 2.4. Testing Predictors of RABV CST Frequency with the Myotis Genus

We assessed the ability of our four host traits, and sample size, to explain variance in the CST frequency between *Myotis* host species using two approaches.

First, we assessed the evidence for a correlation between a matrix of the estimated CST rates between species (MJ count/branch lengths sum/sum of sample sizes of the two species composing a pair) and each of the predictor matrices. We did this by performing mantel tests based on Spearman’s rank correlation values, with 10,000 matrix permutations, using the R package ecodist [44]. Similarly, we also assessed these correlations using multiple regression on matrices (MRM), where predictor variables were removed from the model in reverse order of significance until only significant variables remained. From permutation tests we generated regression coefficients, R-squared values, and F-statistics for lack of fit, along with associated *p*-values.

Secondly, we applied a phylogenetic diffusion approach that simultaneously tests and quantifies potential predictors in a Generalized Linear Model (GLM) framework [42]. Estimated MJ counts, among the fixed number of host species are parameterized as a linear function of one or multiple predictors. We also employed a branch partitioning approach in order to estimate numbers of CST events for the internal and tip branches of the RABV phylogeny separately [19]. This allowed us to calculate the inclusion probability of predictors in two distinct GLMs and to use this to distinguish between predictors of sustained host shifts (internal branches), and un-sustained spill-over (tip branches). BEAST MCMC chains of 800 million steps were run and sub-sampled every 80,000 generations for both analyses. Bayes Factors (BF) – computed by the comparison of the posterior and prior odds that a particular predictor is required to explain the diffusion process – were used as a measure of support for the inclusion of predictors [45]. Support for the inclusion of a predictor was considered substantial when BF > 3, strong if BF > 10, and decisive if >100 [46].

### 2.5. Clustering of Sequences Associated with Myotis Species

In preliminary analyses of the CO1 sequences from the ten *Myotis* host species (details of the host genetic distance matrix building above), we noticed overlap in the within- and between-species host genetic divergences and non-monophyletic clustering in the phylogenetic trees. This is unsurprising given evidence that gene flow exists between some species in this genus [47]. To test whether poorly resolved species impacted our results, we repeated our analysis, employing a hierarchical clustering approach to combine such species into species clusters. Successive application of the R function hclust, the Ward minimum variance clustering method, and Ward’s clustering criterion [48] (which is able to separate clusters even when there is noise between them, and is not biased towards breaking up large clusters) to the previously calculated genetic distance matrix, grouped the sequences into seven clusters. Sequences collected from *M. californicus*, *M. leibii* and *M. ciliolabrum* bats (total *n* = 26) were grouped into one monophyletic group, whilst sequences from *M. keenii* and *M. evotis* (*n* = 38) made up the other monophyletic multi-species cluster. We then repeated the MJ analysis and simultaneous predictor testing using these seven host species clusters as states (Appendix A) in BEAST, as well as the MRM analysis.

## 3. Results

### 3.1. RABV Evolution and CST within North American Myotis Compared to a Multi-Genera Dataset

Based on our set of N gene sequences from *Myotis* bats, RABV was estimated to be evolving at a rate of 2.76 × 10^−4^ (95% HPD interval: 1.83–3.66 × 10^−4^) nucleotide substitutions per site per year, similar to the previously estimated rate for RABV in North American bats overall [19]. The most recent common ancestor (MRCA) of *Myotis* associated bat rabies dated back to 1904 (95% HPD interval: 1866–1941) (Figure 1B). The estimated mean number of CST events over the time period since the MRCA was 40.84 (95% HPD: 34–48). This equates to an expected mean of 3.12 × 10^−4^ CST events per year per capita among all *Myotis* species and, on average, 3.47 × 10^−6^ events per year per capita between each possible pairwise combination of *Myotis* species. The latter estimate was indistinguishable from one we obtained for the independent dataset containing multiple bat genera (3.39 × 10^−6^ events per year per capita) (Figure 2A). However, consistent with our predictions, the proportion of CST events on internal branches was higher in the *Myotis* dataset (on average 46.53%) compared to the multi-genera dataset (on average 37.91%), although the 95% HPD intervals were overlapping (Figure 2B). We also found a greater proportion of species pairs showing evidence of CST in the *Myotis* data compared to the multi-genus dataset (on average 11.96/90 = 13.29% vs. 21.61/272 = 7.95%) with no overlap among 95% HPD intervals (Figure 2C).

The vast majority of inferred CST events among *Myotis* bats were estimated to have occurred in recent decades (Appendix A). Across these inferred CST events between *Myotis* species, some species showed a particular propensity to being the origin or recipient of RABV CST events. The highest CST rates—in descending order—originated from *M. californicus*, *M. evotis*, *M. septentrionalis* and *M. lucifugus*. Meanwhile, *M. lucifugus*, *M. keenii*, *M. evotis* and *M. yumanensis* received the highest number of per year per capita CST events from all other species. RABV moving from and to *M. californicus* and all other species made up 41.7% and 11.8%, respectively, of the inferred CST numbers (Appendix A), a large proportion of which occurred within the last 30 years. CST from and to *M. lucifugus* represented 13.4% and 22.1% of inferred events.

### 3.2. Testing Genetic and Ecological Host Predictors of CST in Myotis Bat Species

Phylogenetic distance between *Myotis* species did not correlate with CST rates in the Mantel test (r = 0.013, *p*-value = 0.45), indicating that virus exchange between species was not constrained by their relatedness. Positive association was seen for species range overlap (r = 0.38, *p*-value = 0.003) but none of the predictor matrices were retained in the model in the MRM analysis. Considering species clusters instead of species, none of the correlations were significant and none of the predictor matrices were retained in the MRM analysis.

We also tested the predictors of the frequency of RABV CST among *Myotis* bat species using a GLM approach in BEAST. Range overlap had a high probability (>0.8) of being included as a predictor in the model and was found to be significant in explaining the number of CST events among the different *Myotis* species (BF = 27.9: strong support) (Figure 3). Similar results were obtained when considering clusters of species specified as discrete states (BF = 5.93: substantial support) (Appendix A). We also found range overlap to significantly correlate with the number of CST events on the internal nodes of the *Myotis* bat RABV phylogeny, representing sustained transmission, i.e., host shifts (BF = 27.6: strong support) (Appendix A). None of the other tested factors were supported as a predictor of CST among *Myotis* (BF < 3), including genetic distance between hosts. Indeed, within the *Myotis* genus, genetic distances were not correlated to the number of CST events per year per capita (correlation coefficient = −0.05, *p*-value = 0.67) whereas a significant negative correlation was found in the dataset with multiple bat genera (correlation coefficient = −0.30, *p*-value = 5.50 × 10^−7^) (Figure 4). In the multi-genera dataset, we inferred especially high numbers of expected CST events from *Lasiurus cinereus* to *Eptesicus fuscus* (mean 7.84 CST events), as well as from *Lasiurus borealis* into *Lasiurus seminolus* (mean 6.79). High numbers of CST also occurred within the *Myotis* genus especially from *M. californicus* to the *M. lucifugus* species complex (mean 6.02 CST events), and from *M. lucifugus* species complex into *Myotis yumanensis* (mean 5.64).

## 4. Discussion

Isolating the factors which influence CST frequency is key to increasing our understanding of pathogen emergence and predicting future outbreaks. Previous studies, both in the lab and the field, have found that the probability of CST and host shifts in RNA viruses decreases with phylogenetic distance between hosts [7,8,19]. Here, we assessed the relative importance of host relatedness and ecological factors in explaining the frequency of CST events among relatively recently diverged species within the same genus. We show that, when host relatedness is high, CST is no longer predictable by host genetic relatedness, but only by the extent of geographic overlap among species.

We found no significant difference between the rates of rabies virus CST within the *Myotis* genus and that of a wider phylogeny of North American bats (Figure 2). This may be because some species pairs in the wider phylogeny exhibit much higher rates of RABV CST than *Myotis* species pairs. Specifically, the exceptionally high rates of CST among *Lasiurus borealis* and *Lasiurus seminolus* raise the multiple genera average as a whole. At the same time, levels of RABV CST for *Myotis* species may be near the maximum level allowed for by ecological opportunity. RABV dynamics within hibernating bats are extremely seasonal, with hibernation thought to maintain RABV infections until the arrival of new susceptibles in the ‘birth pulse’ [49]. The shortness of this transmission window may put an upper limit on the contact rate of infected *Myotis* bats. Indeed, an analysis of substitution rates across bat species found that RABVs associated with temperate (mostly hibernating) bat species evolved more slowly per year, suggesting pauses in transmission which would presumably also reduce CST [50].

When compared with a wider phylogeny of multiple genera of bats, we found some indication for a higher proportion of CST on internal nodes in the *Myotis* RABV phylogeny (Figure 2), though the posteriors overlapped. This suggests that when a virus is passed among host species within the same genus, the proportion of CST events which lead to onward transmission (those found on the internal nodes of the phylogeny), is increased. This is consistent with the idea that viruses are more pre-adapted to establish in novel hosts that are less divergent from the donor host, presumably due to finding a similar physiological and immunological environment [15]. This onward transmission can lead to either a permanent host shift, or the formation of a true multi-species clade of the virus—both of which seem possible in *Myotis*-associated RABV from the viral phylogeny (Figure 1B). Thus, it seems that between closely related host species, it is not the rate of CST which is changed, but the outcome.

We also found that a greater proportion of species within the *Myotis* genus were donors or recipients of CST events compared to North American bats in general, despite the latter group containing other genera with high CST rates (e.g., Lasiurus). In short, when compared with the larger phylogeny, RABV variants infecting more closely related species show similar CST rates, but these CST events involve a greater proportion of host species and are more likely to result in a successful host shift. These observations suggest that between closely related species, fewer barriers to successful RABV host shifts exist, consistent with our predictions.

We also used a phylogenetic diffusion approach to simultaneously reconstruct the ancestral phylogeny of RABV infections within the *Myotis* genus, and to test for possible predictors of the number of CST events along this phylogeny. Previous studies of North American bats, in multiple genera [8,19] found the frequency of both sustained RABV transmission, and spillover, to increase with increased host species relatedness. In contrast, when we implemented this approach within the single genus *Myotis*, we found no significant relationship between host species relatedness and the frequency of CST events. The *Myotis* species have been diverging from their most recent common ancestor for 12.2 ± 2.0 MY, a relatively short period of evolutionary time [51]. Low inter-species divergence could allow for less constrained transmission of RABV variants between species through two non-exclusive mechanisms. First, the shorter divergence time, and resulting similarity in genetic background, between very closely related host species might limit the number of adaptive RABV substitutions needed for one of these variants to invade a new host species and increase the probability of sustained transmission. Standing genetic variation in RABV-infected individuals, in the form of maintained rare variants, would provide the raw material for this adaptation [17]. Indeed, within a single genus, the time between CST and emergence of a new RABV variant has been found to be shorter if fewer positively selected substitutions are needed for RABV variants to invade a new host [52]. Second, closely related host species may also show similar levels of susceptibility to the viral variants due to the loss or gain of common cellular, physiological or immunological components in their phylogenetic history [7]. The evolution of these key determinants of susceptibility may have occurred prior to the diversification of the *Myotis* genus, allowing for less restrained movement of RABV variants. Alternatively, niche partitioning among closely related sympatric species might diminish ecological opportunities for CST despite their higher likelihood of infection after exposure.

Our results suggest that between closely related species, ecological factors become more relevant to explain variation in the frequency of CST than host genetic relatedness. Specifically, we found the geographic range overlap of species to explain a significant proportion of variation in the number and rate of CST events between species (Figure 3). This suggests that at this taxonomic scale, the frequency of CST events is primarily driven by the rate of physical contacts between species. If this is the case, higher rates of CST might be found in regions where a diversity of habitats, and climatic variance, support an increased diversity of host species, as previously suggested for pathogen sharing between primates [53]. In our dataset, this might be the case for the province of British Columbia, which has diverse habitats that support up to nine *Myotis* species [23]. In contrast, we found no consistent evidence that the frequency of CST between *Myotis* bat species is dependent on similarities in their foraging or roosting niches.

It is unlikely that geographic range overlap is the only important ecological factor in explaining CST of RABV among closely related species. However, accurately modelling the complex ecological niches inhabited by bat species is a challenge. For example, we calculated the predictor of foraging niche difference from three morphological measurements as a proxy for niche similarity, a method which may struggle to represent features such as temporal foraging niche differentiation [8]. Additionally, our roosting niche overlap predictor, calculated through the number of overlapping summer roost types, relied on sparse observational literature for some species and might not sufficiently reflect actual ecological interactions to recover a statistical signal. Less social bat species are also often under-represented by passive sampling, hampering the description of their role as RABV reservoirs [54,55]. The recent documentation of the *Myotis* genus supporting its own specific RABV variants [23], further highlights the need for more detailed ecological data from less frequently sampled species.

In this study we investigated the low end of a wide continuum of host genetic divergence and its effect on the probability of a pathogen infecting a novel host. This contrasts with the much larger divergences considered by previous studies of RABV in North American bats. While we show that effects qualitatively differ along this continuum, the specific shape of the relationship between host genetic divergence and CST frequency, which may well be nonlinear, remains elusive. One of the challenges for studying this relationship is the difficulty of meaningfully quantifying the frequency of CST from genetic data given that the detection of events will depend on available sequences and temporal scales captured by the virus phylogeny. However, these limitations are unlikely to qualitatively alter our findings given that neither sample size nor taxonomic classification had a discernible effect on our results. It is also important to emphasise that host divergence or relatedness must be considered proxies for a whole range of host-specific factors that will influence pathogen host range. Ideally, future investigations would seek to replace host relatedness with a more mechanistic way of predicting which viruses are able to establish in a novel host species (e.g., [16]). Ultimately, the relationship between CST and host relatedness will differ between pathogen groups, and will be dependent on the biological mechanisms underlying host specificity in each case. Mapping this relationship as we have done here for a specific group of RABV variants and hosts could thus provide a model for similar studies in other RABV variants and RNA viruses more generally.

## Figures and Tables

**Figure 1 viruses-14-02399-f001:**
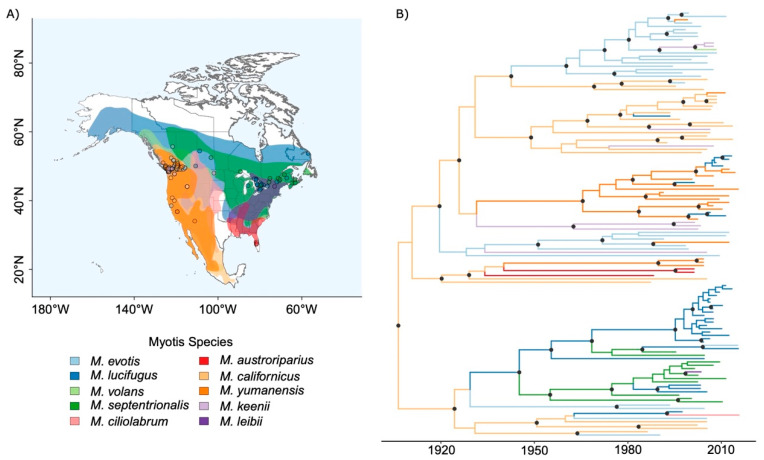
Location and host species origins of samples used in this study (**A**) Map of *Myotis* species sample locations and host ranges, colours indicate host species range and points show the location of collection of a *Myotis* bat infected with rabies virus (RABV). (**B**) Maximum Clade Credibility tree of *Myotis* bat RABV nucleoprotein gene sequences. Branches are coloured by associated host species as specified in the map legend. Node circles indicate nodes with posterior support >0.9.

**Figure 2 viruses-14-02399-f002:**
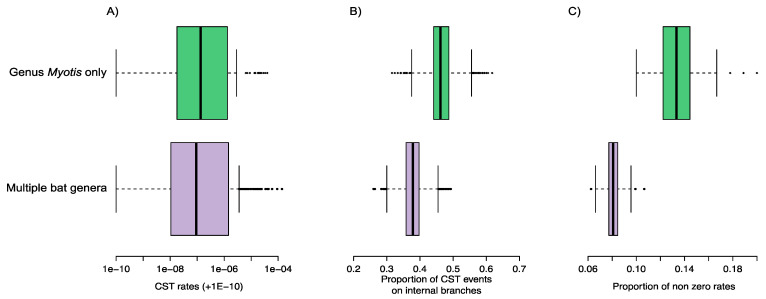
Frequency and characteristics of rabies virus cross-species transmission (CST) for ten North American bat species within the genus *Myotis*, compared to a dataset containing multiple genera of North American bats. Posterior distributions of (**A**) the mean rates (+1E^−10^) of inferred CST events per year per capita for all pairs of species on a log scaled x axis; (**B**) the proportion of CST events on internal branches (indicative of sustained transmission and host shifts); (**C**) the proportion of species pairs with non-zero rates.

**Figure 3 viruses-14-02399-f003:**
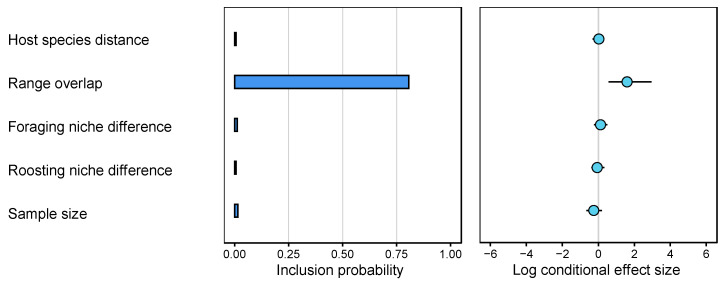
Support and contribution of potential predictors in the phylogenetic Generalized Linear Model (GLM) of rabies CST events within the *Myotis* genus only. For each potential predictor, support is represented by an inclusion probability and a relative contribution indicated for log scale GLM coefficients conditional on the predictor being included in the model (posterior mean and 95% Bayesian confidence interval). Range overlap was the only supported predictor (BF = 27.9).

**Figure 4 viruses-14-02399-f004:**
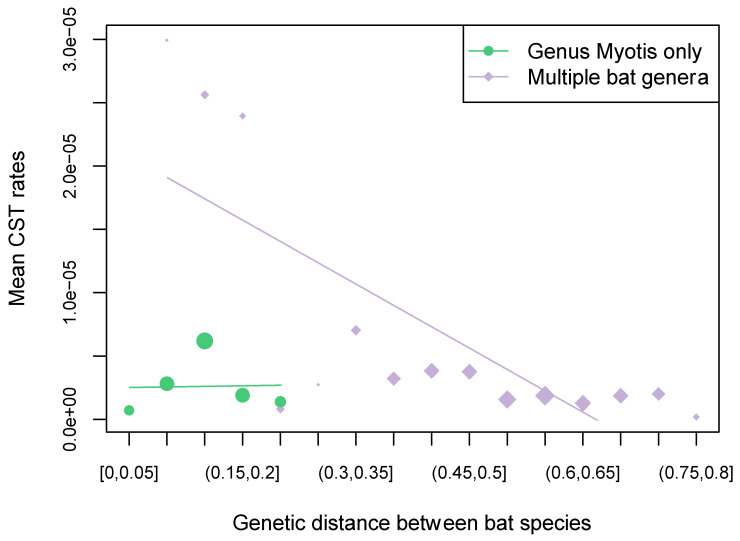
Frequency of inferred cross-species transmission events per year per capita among bat species as function of the genetic distances between species. Intervals of genetic distances values were delimited according to bin widths of 0.5 and means of mean number of expected CST events per year for each bin are shown. Point sizes are log-proportional to sample size (number of pairs of species) within each bin.

## Data Availability

Sample information and GenBank accession numbers of the rabies sequences used in this study are summarized Appendix A. Sources of ecological data used to generate predictors can be found in Appendix A.

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
