# Peer review of "Geographic Range Overlap Rather than Phylogenetic Distance Explains Rabies Virus Transmission among Closely Related Bat Species"

_viruses, 2022, doi:10.3390/v14112399_

Round 1
Reviewer 1 Report
Review Report
ID viruses - 1989582
Title: „Geographic range overlap rather than phylogenetic distance explains rabies virus transmission among closely related bat species„
Author‘s: Maude Jacquot, Megan A. Wallace, Daniel G. Streicker, Roman Biek
Version: 1/ Date: 20-10-2022
Reviewer number: 1
A brief summary
The manuscript „ Geographic range overlap rather than phylogenetic distance explains rabies virus transmission among closely related bat species„ by Maude Jacquot, Megan A. Wallace, Daniel G. Streicker and Roman Biek discusses risk factors for the epidemiologic development and cross-species transmission of rabies viruses in associated bat (Myotis) colonies in unspecified regions of North America. The objective (one main sentence) of the Article is not clearly identified at the end of the “Introduction” part. The other structural parts of the article Methods/ Materials, Results and Discussion…are described clearly and in a sufficiently informative way. The Results are presented in a clear and consistent manner, the tables and figures are informative (although some Figs. are poorly designed) and the statistics (basic) are presented in the text of the Results. The results of the research are relatively compared with the data of other scientists in the Discussion section, elements of comparative analyses and personal opinion can be found.
Broad comments
The materials of the article may be of great interest to a wide range of scientists who work with rabies epidemiological data in wild fauna and directly with the development of rabies in bat populations. The scientific discussion of the article can be especially valuable in the context of cross-species transmission of rabies viruses in wildlife and in assessing the risk potential of human co-infection.
The Abstract needs to be reviewed: the first sentences (up to line (L) 28…” Previous work …”) are of general interest and are not closely related to the tasks of the article. The research results and conclusions should be emphasized in the abstract of the scientific article. In the classical Summary should (not must) contain the purpose (objective) of the article (one sentence), 2-3 main tasks (2-3 sentences), the results (2-3 sentences) and conclusion (one sentence again).
The Introduction prepared well (in advance) with possibly focus on the research topics of study, but it just needs to be repeated – the objective (one main sentence) is not clearly identified at the end of this part (remember the strategic definitions of the goal and tasks).
The Materials and Methods described in detail, although the criteria for the choice of methods could be detailed according to the work tasks. Statistical analysis additional information should be provided in this section – the selection criteria for groups comparison and the calculation of the correlation (between and within).
The Results are well presented and sufficiently understandable.
All the presented research results should be discussed in the article. The most interesting results scientific-based) should be discussed on a priority basis, if it is not possible to compare such results with other authors data, the personal interpretations are strongly recommended.
The list of reference should be revised and unified according to the specific recommendation for authors.
Specific comments
There are some specific recommendations based on my personal opinion:
L 2-4. Title. North America and Myotis should be added…
L 18-19. Keywords: rabies, cross-species transmission, Myotis bat, North America
L 34. Insert RABV before ,,…sequences…,,
L 49-50. ,,(Andersen et al., 2020; Morens et al., 2008; …) ,,…and see reviews…,, should be deleted.
L 53. ,,(eg..),, should be deleted as well.
L 64. ,,(see..) should be deleted as well.
L 81. ,,(… and see review..) should be deleted as well.
L. 95-96, 97, 161…. Plain text, family / subfamily names in Italic Myotis, Vespertilionidae, Myotinae
L. 94-107. …looks like some results, you need to rearrange the information and form one goal and several tasks of your research.
L. 131, 146, 147… The first-time abbreviation must be explained (Reffer) in the text (BEAST, MCMC, BEAGLE…) – important if it is read by a ,,unspecialized,, reader.
L. 279-280. ..originated, received – Plain text.
L . 355-356. An inappropriate ,,parallels,, - a different RABV biological and ecological cycle, a different model of social development…
Additionally. If statistical calculations are performed - it is necessary to detail in Results and Discussion what additional value these data provide and what their interpretive significance is in scientific analysis (why that statistics was used and what it results means). If no data available - no discussion presentable. ,,et al,,, or ,,et al.,,,? All the authors must be identified in References sources. The list of abbreviations should be prepared and added.

Author Response
-
A brief summary - The manuscript „ Geographic range overlap rather than phylogenetic distance explains rabies virus transmission among closely related bat species„ by Maude Jacquot, Megan A. Wallace, Daniel G. Streicker and Roman Biek discusses risk factors for the epidemiologic development and cross-species transmission of rabies viruses in associated bat (Myotis) colonies in unspecified regions of North America. The objective (one main sentence) of the Article is not clearly identified at the end of the “Introduction” part.
RE: We have added a sentence to the beginning of the last paragraph of the introduction outlining our primary objective of the study - ‘Here, our primary aim was to ask if high host relatedness can still increase the probability of a host shift between more recently diverged hosts, and the importance of this effect relative to ecological predictors. To do this we….’ – which we hope makes this section clearer for the reader.
-
The other structural parts of the article Methods/ Materials, Results and Discussion…are described clearly and in a sufficiently informative way. The Results are presented in a clear and consistent manner, the tables and figures are informative (although some Figs. are poorly designed) and the statistics (basic) are presented in the text of the Results. The results of the research are relatively compared with the data of other scientists in the Discussion section, elements of comparative analyses and personal opinion can be found.
-
Broad comments - The materials of the article may be of great interest to a wide range of scientists who work with rabies epidemiological data in wild fauna and directly with the development of rabies in bat populations. The scientific discussion of the article can be especially valuable in the context ofcross-species transmission of rabies viruses in wildlife and in assessing the risk potential of human co-infection.
RE: We thank reviewer 1 for these positive comments, and have updated figure 4 to a higher quality version and using darker colours (same colours as Fig 2), as it was also referenced by reviewer 2. We believe this improve readability of the figure.
-
The Abstract needs to be reviewed: the first sentences (up to line (L) 28…” Previous work …”) are of general interest and are not closely related to the tasks of the article. The research results and conclusions should be emphasized in the abstract of the scientific article. In the classical Summary should (not must) contain the purpose (objective) of the article (one sentence), 2-3 main tasks (2-3 sentences), the results (2-3 sentences) and conclusion (one sentence again).
RE: We feel that the first two sentences of the abstract provide background and, as specified in the author instructions for the journal, ‘Place the question addressed in a broad context’ and so have not removed them, but have made them more concise. We have tried to make the rest of the abstract structure – particularly in terms of outlining the main aim and tasks of the research – clearer by re-organising it.
-
The Introduction prepared well (in advance) with possibly focus on the research topics of study, but it just needs to be repeated – the objective (one main sentence) is not clearly identified at the end of this part (remember the strategic definitions of the goal and tasks).
RE: We have added a sentence to the beginning of the last paragraph of the introduction outlining our primary objective of the study – see details above.
-
The Materials and Methods described in detail, although the criteria for the choice of methods could be detailed according to the work tasks. Statistical analysis additional information should be provided in this section – the selection criteria for groups comparison and the calculation of the correlation (between and within).
RE: We are sorry, we did not understand to which statistical analysis reviewer 1 was referring to. The only “groups” we defined in our study are based on the Myotis species clustering to which a full section of the materiel and methods section is dedicated “Clustering of sequences associated with Myotis species“. However, to clarify this section we have added some more details on why we chose the particular clustering method we used, and where the genetic distance matrix this is applied to came from.
-
The Results are well presented and sufficiently understandable. All the presented research results should be discussed in the article. The most interesting results scientific-based) should be discussed on a priority basis, if it is not possible to compare such results with other authors data, the personal interpretations are strongly recommended.
RE: Thank you for raising this point. We believe, as you suggest, that all presented results are well described in the manuscript and discussed in the light of current literature and our expertise.
-
The list of reference should be revised and unified according to the specific recommendation for authors.
RE: Done.
-
Specific comments- There are some specific recommendations based on my personal opinion:
L 2-4. Title. North America and Myotis should be added…
RE: Thank you for this suggestion. While we think it might be relevant to add this to the title, we also believe it makes the title too long and therefore decided not to modify it.
L 18-19. Keywords: rabies, cross-species transmission, Myotis bat, North America
RE: Missing keywords added.
L 34. Insert RABV before ,,…sequences…,,
RE: Done.
L 49-50. ,,(Andersen et al., 2020; Morens et al., 2008; …) ,,…and see reviews…,, should be deleted.
L 53. ,,(eg..),, should be deleted as well.
L 64. ,,(see..) should be deleted as well.
L 81. ,,(… and see review..) should be deleted as well.
RE: Done.
-
95-96, 97, 161…. Plain text, family / subfamily names in Italic Myotis, Vespertilionidae, Myotinae
RE: Done.
-
94-107. …looks like some results, you need to rearrange the information and form one goal and several tasks of your research.
RE: We have added a sentence to the beginning of the last paragraph of the introduction outlining our primary objective of the study, and additionally, have altered some of the wording and order of the sentences later in the paragraph to make the tasks of the research clearer, and match up with the tasks defined in the abstract.
-
131, 146, 147… The first-time abbreviation must be explained (Reffer) in the text (BEAST, MCMC, BEAGLE…) – important if it is read by a ,,unspecialized,, reader.
RE: We agree that these abbreviations need defined for unspecialised readers, and have added them into the text on their first use.
-
279-280. ..originated, received – Plain text.
RE : We had originally italicised these words for emphasis, but agree that this could be confusing given the other italicised words in the sentence, so have changed them to plain text
L . 355-356. An inappropriate ,,parallels,, - a different RABV biological and ecological cycle, a different model of social development…
RE : We have removed this comparison from the text.
-
If statistical calculations are performed - it is necessary to detail in Results and Discussion what additional value these data provide and what their interpretive significance is in scientific analysis (why that statistics was used and what it results means). If no data available - no discussion presentable. ,,et al,,, or ,,et al.,,,?
RE: We believe this was already done in the former version of the manuscript. No additional change has been made.
-
All the authors must be identified in References sources.
RE: The reference list has been updated in its formatting according to the instructions for authors, and so all authors are now identified.
-
The list of abbreviations should be prepared and added.
RE: With reference to the instructions for authors, we have checked that all abbreviations are defined the first time they appear in each of the three sections of the text (abstract, main text and first figure or table) – including some definitions added to the legend of figure 3 for this purpose.

Reviewer 2 Report
The authors should check the journal's instructions to adapt the formatting of the manuscript, especially with regard to in-text citations and references.
Some comments were added to the pdf file.

Author Response
-
L47: References must be numbered in order of appearance in the text. In the text, reference numbers should be placed in square brackets [ ], and placed before the punctuation; for example [1], [1–3] or [1,3]
-
L49: All the citations must be corrected as indicated in the comment above.
RE: Done, also referred to above.
-
L95: remove the italics
RE: Done.
-
L319, Figure 4: The quality of the figure must be improved. It's no so clear.
RE: Done.
-
L449, References. Correct the references according to instructions to authors.
RE: Done, also referred to above.
